

# Attention-based BiLSTM-XGBoost model for reliability assessment and lifetime prediction of digital microfluidic systems

Lifeng He*, Qili Yang*, Junxi Chen, Wenjing Liu and Zhijie Luo

Zhongkai University of Agriculture and Engineering, College of Information Science and Technology, Guangzhou, Guangdong Province, China
* These authors contributed equally to this work.

## ABSTRACT

Traditional methods for reliability and lifetime testing of digital microfluidic systems heavily rely on real-time monitoring data. This often leads to evaluation lag and limits their application, especially for complex droplets. To address these issues, this study proposes a novel prediction model for digital microfluidic (DMF) devices. The model combines an attention-based bidirectional long short-term memory (BiLSTM) with eXtreme Gradient Boosting (XGBoost) using a Stacking approach. This integrated model efficiently identifies the health state and predicts the failure time of digital microfluidic devices. This approach overcomes the limitations of traditional methods, such as over-reliance on sensor feedback and detection hysteresis. Experimental results demonstrate high prediction accuracy. The model achieved a mean absolute percentage error (MAPE) of 1.6464, Root mean squared error (RMSE) of 0.3667, mean absolute error (MAE) of 0.2557, and a coefficient of determination (R-squared) of 0.9949. Compared to baseline methods, the proposed BiLSTM-XGBoost model achieves the highest prediction accuracy, enabling effective health monitoring, problem identification, and failure prediction. Ultimately, this improves system reliability and lifetime with greater timeliness and accuracy.

## INTRODUCTION

Digital microfluidic (DMF) systems based on electrowetting-on-dielectric (EWOD) have emerged as a promising platform in biomedical research, laboratory automation, and medical diagnostics. With these devices, researchers can manipulate micro-scale droplets and biological samples efficiently and precisely. This enables a wide range of experimental applications in disciplines such as biology, chemistry, and medicine.

In recent years, significant advances have been made in both the theoretical modeling and practical implementation of DMF systems. *Tong et al. (2023)* proposed a DMF device with integrated sensors and actuators designed for biomedical applications. *Torabinia et al. (2021)* developed an efficient EWOD platform for synthetic organic chemistry, while *Sagar, Bansal & Sen (2022)* introduced an open-type EWOD device with enhanced operational flexibility.

Corresponding author
Zhijie Luo, jackeylzj@163.com

Despite recent advancements, assessing DMF device reliability and predicting their lifetime remain significant challenges. Current global studies largely focus on real-time or offline detection of driving electrode states (*Ghosh, Roy & Giri, 2021*; *Howladar, Roy & Rahaman, 2020*). However, systematic long-term research on DMF system reliability and degradation mechanisms is still lacking. As DMF is sensitive and used in safety-critical point-of-care diagnostics, ensuring robustness and enabling predictive maintenance is essential. Traditional monitoring fails to capture complex temporal dynamics and nonlinear dependencies. Thus, advanced data-driven models are necessary.

It is worth noting that some relatively mature reliability assessments (*Li et al., 2021*; *Bian et al., 2023*; *Li & Wang, 2022*) and remaining life prediction methods already exist in other fields (*Xiao et al., 2023*; *Wei, Wu & Terpenny, 2024*; *Sekhar, Domathoti & Santibanez Gonzalez, 2023*). Deep learning has been increasingly applied to reliability evaluation and remaining life prediction in other domains, leading to various techniques that have drawn attention (*Zhang et al., 2019*; *Pan et al., 2025*; *Kumar, Khalid & Kim, 2023*). However, the direct applicability of these methods to DMF systems remains uncertain. Unlike mechanical or large-scale systems, DMF devices operate on a microscale. This introduces different physical constraints, including unique droplet actuation mechanisms and sensitivity to contamination. Furthermore, fault observability is often limited. Therefore, direct adoption of such methods without domain-specific validation may not yield reliable results.

Recognizing the need for a domain-specific solution, this study proposes a reliability and lifetime prediction framework for DMF systems. A hybrid Attention-based bidirectional long short-term memory with eXtreme Gradient Boosting (BiLSTM-XGBoost) model is used for this purpose. The method is designed to capture the long-term trends of the equivalent capacitance values of the driving electrodes. This enables real-time health monitoring and facilitates proactive adjustment of control strategies based on predictive insights. The proposed approach leverages bidirectional long short-term memory's (BiLSTM) temporal modeling, an attention mechanism, and eXtreme Gradient Boosting's (XGBoost) classification strength. This enhances the accuracy and interpretability of system state assessments. This research offers a novel pathway for improving DMF system operational reliability. Its insights could also be extended to fault prediction in other complex microsystems.

## DRIVING AND DETECTION MECHANISM OF DMF SYSTEM BASED ON EWOD

At this stage, two main chip structures exist for DMF systems based on EWOD: closed and open. In the open structure, the droplet is directly placed on a single substrate integrating both drive and ground electrodes. This design can simplify integration and fabrication. However, open systems present challenges: droplets are more vulnerable to environmental influences during operation, and complex multi-droplet manipulation, such as splitting and generation, is harder to achieve (*Wang & Jones, 2015*). Therefore, more research teams focus on the closed structure, which generally offers enhanced control and functionality for complex droplet operations (*Abdelgawad, 2020*; *Xing et al., 2021*).

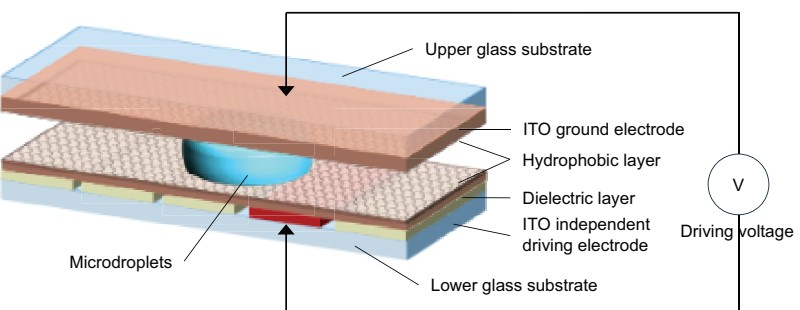

**Figure 1 Schematic diagram of the application mechanism of DMF system with bipolar plate closure.**               

Figure 1 shows a schematic of a bipolar plate-enclosed DMF chip. In this structure, the top substrate is a continuous transparent conductive indium tin oxide (ITO) layer that acts as the ground electrode. The bottom substrate contains patterned drive electrodes fabricated by photolithography, typically in a square shape. Both the upper and lower substrates are coated with a dielectric layer and a hydrophobic layer. After device fabrication, droplets are injected into the space between the two substrates. While air is usually used as the surrounding medium, some teams fill this space with a liquid immiscible with the droplet as a lubricant to reduce the required driving voltage.

When the drive electrode is not activated, the droplet remains at rest in an ellipsoid shape. Once a neighboring electrode is activated, the solid–liquid interfacial energy decreases. This leads to a smaller contact angle and an asymmetric droplet shape, creating a pressure difference that drives the droplet toward the activated electrode.

According to the principle of electrowetting, capacitance is a key property in EWOD systems (*Luo et al., 2018*). The bipolar plate DMF chip can be modeled as an equivalent capacitive system. For a basic driving unit, the equivalent circuit includes three main parallel capacitance components. First, the dielectric and hydrophobic layers on the bottom plate form one equivalent capacitance. Second, the hydrophobic layers on both plates in contact with the droplet form another, but with higher capacitance. In a series configuration, the voltage drop across the upper layer's hydrophobic capacitance is minimal. Most of the voltage drops across the lower layer. Thus, in the equivalent model, the droplet serves as the ground, and the surrounding medium forms a capacitor.

As the electrode surface degrades, the equivalent capacitance decreases. Severely damaged may cause capacitance to drop sharply. By monitoring the capacitance at both ends of a drive electrode, the electrode's health status can be assessed.

Based on more than 100,000 droplet-driving operations across 20+ EWOD devices, we classify the degradation into three stages. When the equivalent capacitance remains at 80–100% of its original value, the droplet can usually be transported successfully in a single attempt, with only 2–3 V above the initial driving threshold. This is the normal degradation period. When the capacitance drops to 50–80%, the droplet often requires 3–5 repeated driving attempts or an increase of 3–5 V to move successfully. This is the recession period. When the capacitance is below 50%, a much higher voltage is needed. In

many cases, the droplet becomes stuck on the electrode and fails to move. This stage is defined as the damage period.

# RELIABILITY ASSESSMENT AND FAILURE TIME PREDICTION MODEL BASED ON BILSTM-XGBOOST MODEL

## Modelling composition

The reliability assessment and failure time prediction model in this study is based on a BiLSTM-XGBoost architecture. It combines BiLSTM networks (*Zhang et al., 2015*) with the XGBoost algorithm (*Chen & Guestrin, 2016*). The goal is to leverage the bidirectional sequence modeling capability of BiLSTM and the efficient learning performance of XGBoost to improve the overall accuracy and robustness of predictions. Specifically, the time-series features extracted by the BiLSTM are further processed by XGBoost to enhance the model's ability to analyze temporal patterns.

Long short-term memory (LSTM) is a specialized type of recurrent neural network (RNN) designed to address the vanishing and exploding gradient problems. It can effectively capture long-term dependencies in time-series data (*Hochreiter & Schmidhuber, 1997*). In the context of health state recognition and failure time prediction, LSTM networks are widely used to extract time-dependent features from operational data and to track temporal changes (*Nguyen et al., 2021; Zhang et al., 2020; Yang & Kim, 2018*).

An LSTM architecture can be represented as the architecture in Fig. 2, where $H_n$ is the output value or hidden state, $C_n$ is the current value of the storage unit and $X_n$ is the input value.

The input sequence of an LSTM can be shown as $\{x_1, x_2, \ldots, x_t\}$, where the subsequent $x_t \in R^K$ is a k-dimensional vector associated with the $t$-th time interval. In LSTM, the equation for the gate is

$$f_t = \sigma\left(W_f[h_{t-1}, x_t] + b_f\right) \tag{1}$$
$$i_t = \sigma\left(W_i[h_{t-1}, x_t] + b_i\right) \tag{2}$$
$$O_t = \sigma\left(W_o[h_{t-1}, x_t] + b_o\right). \tag{3}$$

Here, the forgetting gate $f_t$, $i_t$ and $O_t$ are the forget, input, and output gates, respectively. These gates regulate the flow of information into and out of the memory cell. The forget gate decides which information from the previous state should be discarded. The input gate determines which new information to store. The output gate controls what information to pass on to the next layer or output. The parameters (weights and biases) governing these gates are learned automatically during the model training process.

The cell state is updated using a candidate value $\tilde{C}_t$ computed as:

$$\tilde{C}_t = \tanh(W_c[h_{t-1}, x_t] + b_c) \tag{4}$$
$$C_t = f_t C_{t-1} + i_t \tilde{C}_t \tag{5}$$
$$h_t = O_t tanh(C'_t) \tag{6}$$

where $W_c$ is a candidate vector weight; $b_c$ is a candidate vector deviation; $C_t$ is a current candidate vector is the updated value of the candidate vector at time $t$.

**Peer**J Computer Science

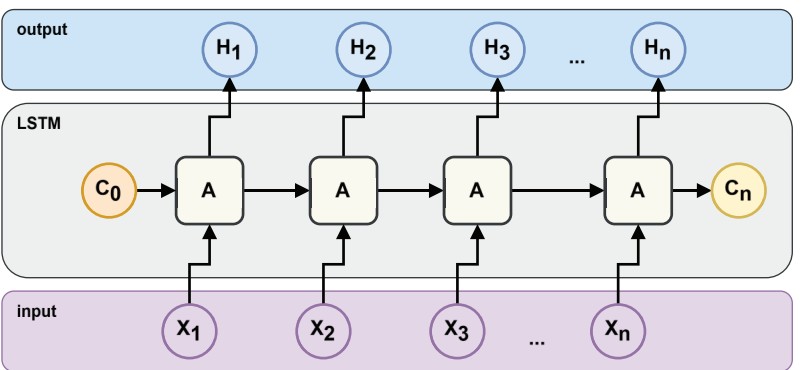

**Figure 2  LSTM neural network structure.**

Meanwhile, the candidate vector ($C_t$) plays a crucial role in capturing and updating information. It is computed by combining the current input ($C_t$) and the previous hidden state ($h_{t-1}$) using the candidate vector weights ($W_c$) and a deviation term ($b_c$). The candidate vectors represent information that can be merged into the cell state. By applying an activation function to the candidate vector, an updated value of the candidate vector is obtained, denoted as $\tilde{C}_t$. This updated value helps to compute the new cell state and affects the flow of information within the LSTM cell. The candidate vector weights ($W_c$) and the bias term ($b_c$) provide parameters for the model to learn how to integrate the new input information with the existing state.

However, traditional RNNs, including standard LSTMs, process information only in the forward direction. However, in many prediction tasks, future context is also valuable. Allowing the model to learn the prediction task through both future and past information can get more powerful features to improve the model's generalization in the prediction task. To incorporate both past and future information, *Schuster & Paliwal (1997)* proposed the bidirectional RNN. *Graves & Schmidhuber (2005)* extended this idea to LSTM, resulting in the BiLSTM architecture (see Fig. 3), which combines forward and backward LSTM outputs to generate a richer representation of the input sequence.

To further enhance the model's ability to capture contextual relationships, we introduce an attention mechanism. The attention layer allows the model to selectively focus on important parts of the sequence. Originally proposed for computer vision (*Mnih et al., 2014*), the attention mechanism has since been widely adopted in natural language processing (*Bahdanau, Cho & Bengio, 2014*; *Luong, Pham & Manning, 2015*) and time-series forecasting tasks (*Lu et al., 2024*; *Che et al., 2024*). As illustrated in Fig. 4, the attention layer assigns dynamic weights to different input elements, enabling the model to prioritize relevant temporal features.

XGBoost is a powerful decision-tree-based algorithm designed for classification and regression tasks. Its fast learning speed and strong generalization make it an effective complement to neural networks. XGBoost enhances the overall prediction performance without significantly increasing computational cost, and has demonstrated strong results across various domains (*Zhang, Jia & Shang, 2022*; *Yang et al., 2023*).

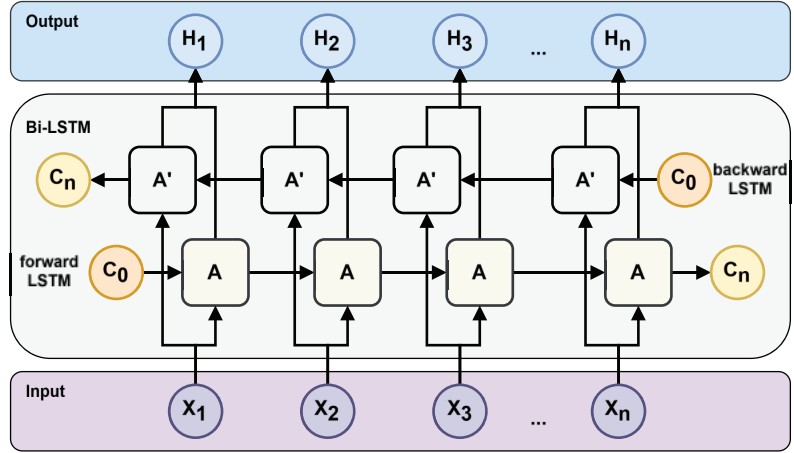

**Figure 3  Bi-LSTM neural network structure.**

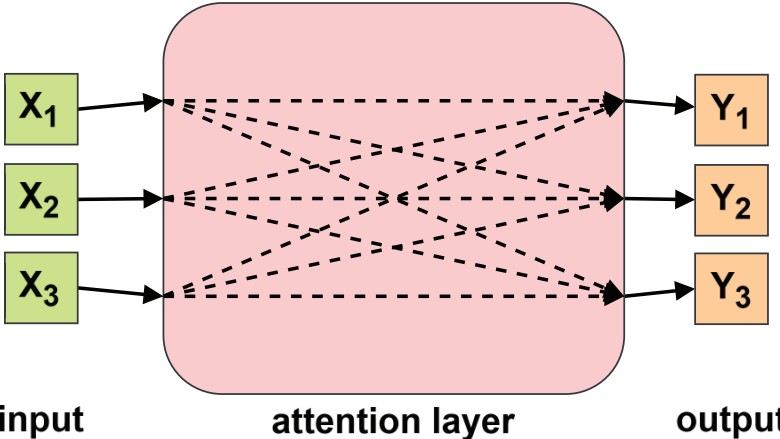

**Figure 4  Attention mechanism.**

Many recent studies (*Lou et al., 2024*; *Javeed et al., 2023*; *Chang et al., 2023*) have shown that BiLSTM-XGBoost models consistently outperform single-model approaches in both accuracy and robustness. Therefore, in this work, we adopt the BiLSTM-XGBoost architecture to analyze the time-series capacitance data of EWOD-based DMF devices, aiming to evaluate reliability and predict remaining useful life (RUL).

## Forecasting process

The state evaluation process of the DMF device is shown in Fig. 5.

The following steps are mainly included in the evaluation process:

1. Data preprocessing: firstly, real-time capacitance values of DMF devices are collected, and the data are subjected to preprocessing operations such as data cleaning, denoising, and normalization to ensure the quality and consistency of the data.

   We first use the weighted moving average filtering method to remove the high and low-frequency noise in the original time domain data to improve the smoothness and

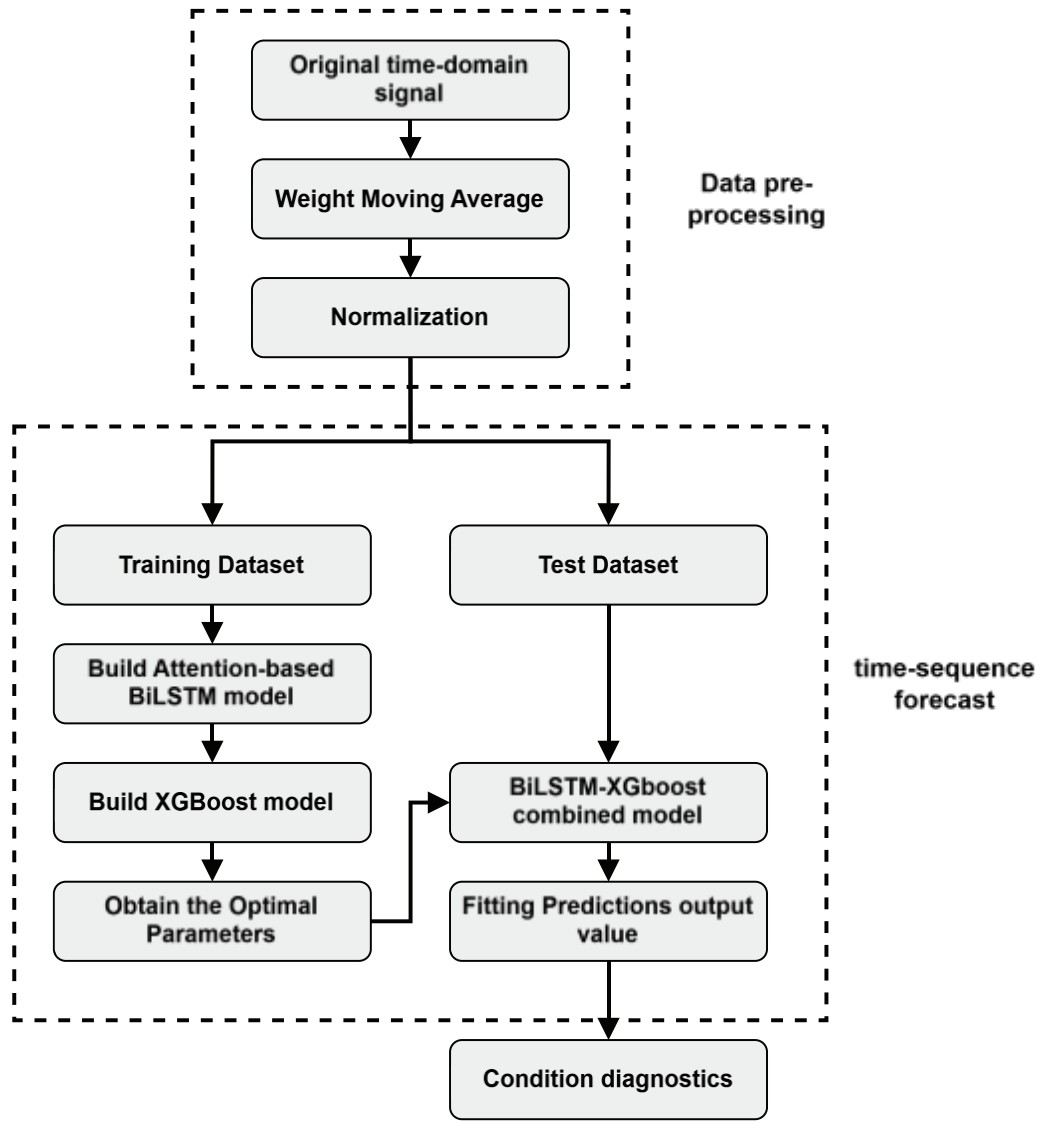

**Figure 5 Health state assessment process for DMF devices.**

accuracy of the data, thus accelerating the speed of model convergence. The weighted sliding average filtering method is shown in Eq. (7).

$$\overline{y_i} = \frac{\left(\sum_{k=1}^{N} w_k y_{(i+k)}\right)}{\left(\sum_{k=1}^{N} w_k\right)}. \tag{7}$$

In Eq. (7), $y_i$ is the original time-domain data, $\overline{y_i}$ is the filtered data; $w_k$ is the linear weight coefficient, whose expression is shown in Eq. (8).

$$w_k = N + 1 - K. \tag{8}$$

In order to eliminate the interference of magnitude and order of magnitude on the model training results, and at the same time speed up the model convergence, we need to normalize the filtered data. The normalization process is shown in Eq. (9).

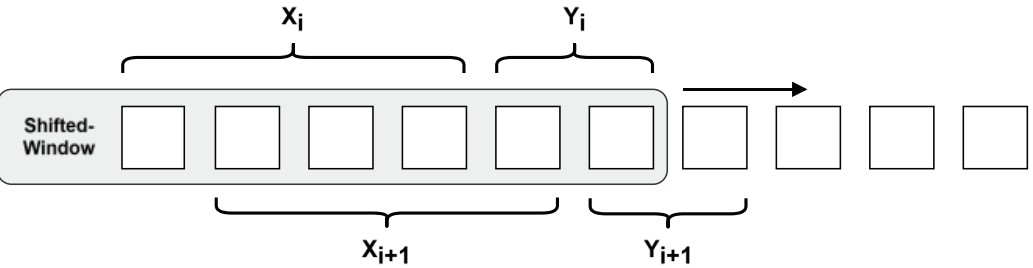

**Figure 6** Shifted-window methods for constructing RNN data structures.

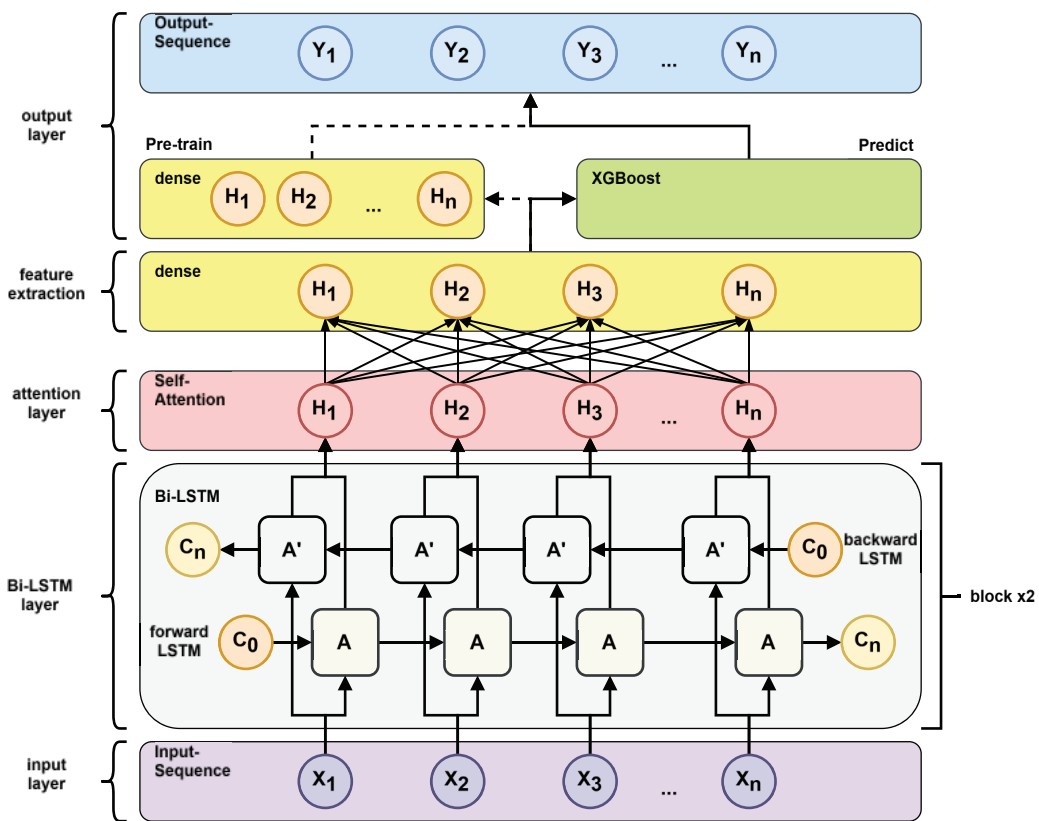

**Figure 7** Pre-train *vs.* predict process.

$$\hat{x} = \frac{x - x_{\min}}{x_{\max} - x_{\min}}. \tag{9}$$

In Eq. (9), $\hat{x}$ is the normalized data, while $x$ is the original monitoring data, $x_{\max}$ and $x_{\min}$ are the maximum and minimum values of the monitoring data, respectively.

To cope with the demand of predicting long-step time series and avoid the model using shortcut learning to fit the curve, a more challenging pre-training task needs to be constructed. As shown in Fig. 6, a shifted-window approach is used to construct the RNN

data structure, which slices the time series into a set of 50 data points as input, and this is used to predict the latter set of time series with a step size of 25.

2. Model training: as shown in Fig. 7 after inputting the training set to the BiLSTM model in the pre-training stage, key time series features are extracted to derive the short-term and long-term dependence of the capacitance values of the DMF devices at different times and the extracted features are transformed into the time series to be predicted by the multilayer perceptron (MLP). After pre-training the BiLSTM model, the fully connected layer used to fit the time series is removed and the data features learned by the BiLSTM in the training set are fed into the XGBoost model for further fitting and the output of the XGBoost model is used as the final prediction result.

3. Time series prediction: Use the BiLSTM-XGBoost model to predict the future trend of the capacitor.

4. Reliability assessment and remaining life prediction: Based on the results predicted by the BiLSTM-XGBoost model, the DMF devices are assessed for reliability and predicted for failure time. The current health status of the device is judged by analyzing the sequence characteristics of the predicted values and combining them with the definition of the device's operating state. If the device is in the decline phase of its life cycle, the remaining life of the device is predicted.

## EXPERIMENTS AND ANALYSES

### DMF device fabrication for experiments

In this study, we fabricated the EWOD-based DMF devices required for this research in a clean laboratory using micromechanical technology processes. To simplify the process and time for fabricating the chip, we spin-coated 1,200 nm thick Teflon AF1600 material on the surface of the ITO layer of the upper and lower pole plates of the device, which serves as both the dielectric and hydrophobic layers of the device. We used SU-8 photoresist as a support wall for the top and bottom polar plates of the EWOD device to keep enough space for droplet movement between the top and bottom polar plates of the EWOD device. The structural parameters of the EWOD device used for the experiments in this study are shown in Table 1. The droplet volume is about six μL, and the horizontal diameter of the droplet on the EWOD device is about 3.5 mm. The DMF device and the external drive control system are connected and communicated through a piece of PCB copper foil. Each drive electrode can be controlled by a separate drive control port (which can output 30–60 V).

### Data set description and analysis

In this study, experimental data comprising capacitance values from the EWOD device were collected using the equivalent capacitance acquisition system previously proposed by the research team (*Luo et al., 2018*). This dataset is publicly available on Figshare at https://doi.org/10.6084/m9.figshare.28941695. To collect this data and

**Table 1 Parameters of DMF devices for experiments.**

| Device parameters | Value |
|---|---|
| Drive electrode shape | Square |
| Drive electrode size | 3 mm |
| Drive electrode pitch | 50 μm |
| Distance between the upper and lower plates of the device | 1.5 mm |
| Diameter of vertical surface of droplet in static state | 3.5 mm |

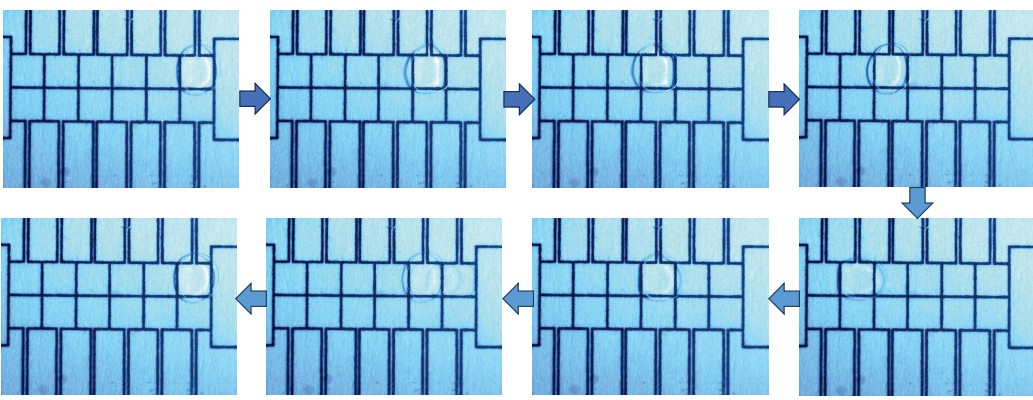

**Figure 8 The droplet travels back and forth between the two driving electrodes.**

investigate the model's performance under various operating conditions, different driving voltages (45, 70, 90 V) were applied to actuate the droplets back and forth between the two driving electrodes on the EWOD device shown in Fig. 8.

The system collects and records the current capacitance value after each drive. Each driving voltage drove the droplet 2,000 times (1,000 times for each of the two electrodes) on three completely independent EWOD devices, and a total of 1,000 capacitance value data were obtained for each group, for a total of 6 groups. The equivalent capacitance values of 18 groups (45, 70, 90 V) of driving electrodes were finally obtained as experimental data input to the proposed prediction model, as shown in Fig. 9, and for the six groups of data for each voltage, the number of driving times as the $x$-axis and the capacitance values as the $y$-axis is labeled as data1–data6, respectively.

Figure 9 shows that the capacitance value of the drive electrode decreases overall with the increase of the number of drives in multiple experiments, and the multiple drives lead to the aging and depletion of the capacitance of the drive electrode, and the decrease of the capacitance value reflects the gradual decrease of the reliability of the DMF system with the increase of the operation time. For different driving voltages, due to the inconsistent trend of device performance degradation, it is necessary to conduct independent experiments for different driving voltages separately and examine the prediction performance of different prediction models under different driving voltages, to determine the optimal prediction scheme.

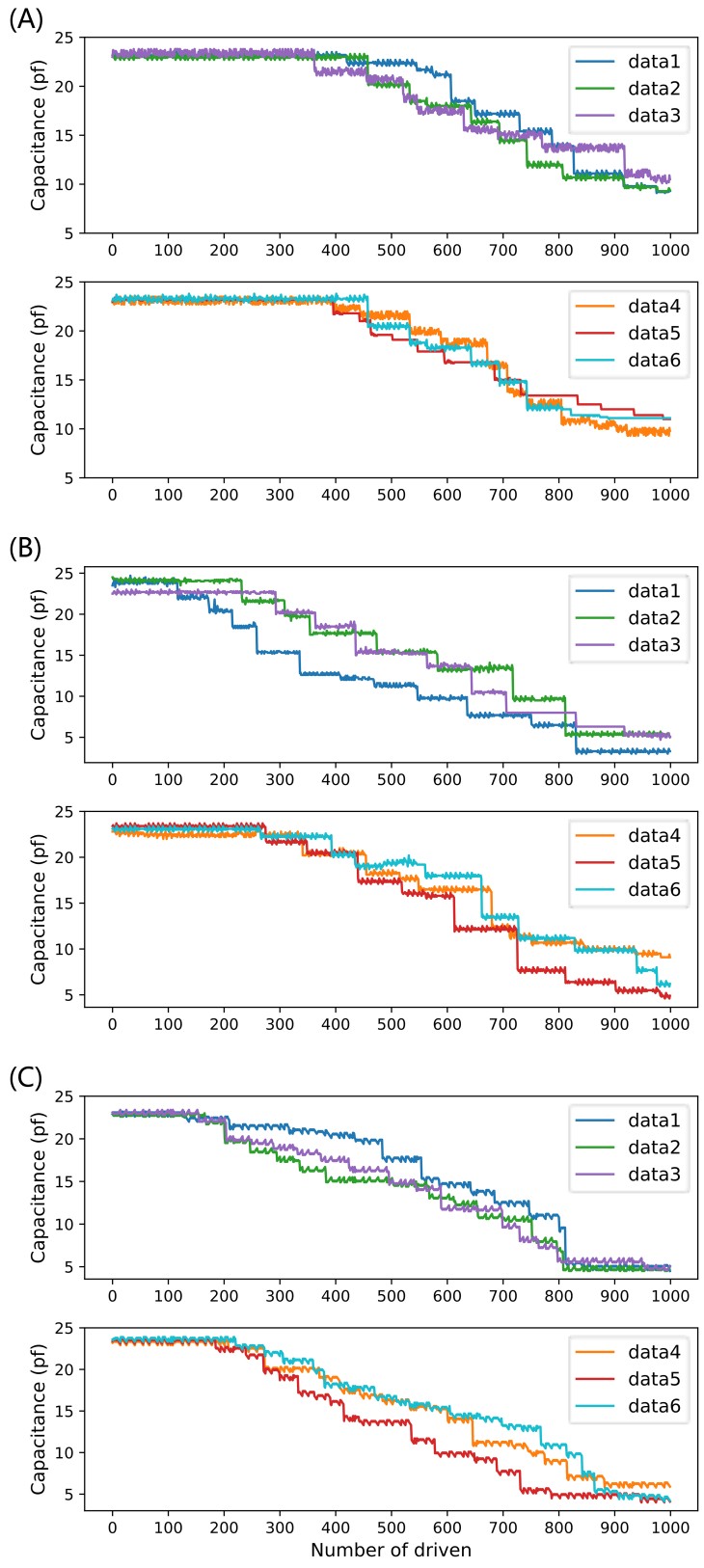

**Figure 9 Six sets of full-life capacitance variation data at three driving voltages: (A) 45 V driving voltage; (B) 70 V driving voltage; (C) 90 V drive voltage.**

**Table 2 RMSE of the three optimization algorithms.**

| Optimizer | Adam | SGDM | RMSprop |
|---|---|---|---|
| First experiment | 0.5027 | 1.2362 | 0.5285 |
| Second experiment | 0.5192 | 1.2329 | 0.5317 |
| Third experiment | 0.4936 | 1.2196 | 0.5178 |
| Fourth experiment | 0.5560 | 1.2667 | 0.5238 |
| Fifth experiment | 0.4399 | 1.2631 | 0.5050 |

Note:
Adaptive Moment Estimation (Adam); stochastic gradient descent with momentum (SGDM); root mean square propagation (RMSprop).

In the time series prediction stage, we select the time series data of the capacitance value as the sample input to predict the future trend of the capacitance value of the device.

## Model training

For each driving voltage, we select the first four of the six experimentally measured full-life data (data1–data4) as the training set, the fifth data (data5) as the cross-validation set, and the sixth data (data6) as the test set. After dividing the training, cross-validation, and test sets, the original time-domain signals are denoised and normalized according to the method in "Data Set Description and Analysis". The RNN model data structure is constructed. Then, the processed training set is input into the BiLSTM model for training, where the hyperparameters of the BiLSTM model are iterated by the cross-validation set using the grid search method to find the optimal combination.

In the model training phase, three commonly used model weight optimization algorithms are Adaptive Moment Estimation (ADAM) (*Kingma & Ba, 2014*), stochastic gradient descent with momentum (SGDM), and root mean square propagation (RMSprop), in this study, we use these three optimization algorithms to conduct five experiments respectively, and the results of the root mean square error (RMSE) of the three optimization algorithms are shown in Table 2. Among the three optimization algorithms tested (SGDM, Adam, and RMSprop), Adam generally yielded the smallest RMSE across our experiments. Therefore, we used the Adam optimizer for gradient optimization to obtain the optimal model parameters in this experiment. Therefore, in this experiment, we use the Adam optimizer for gradient optimization to obtain the optimal model parameters it can be seen that among the three optimization algorithms, the RMSE of Adam and RMSprop is significantly smaller than that of SGDM. Although Adam and RMSprop have their own advantages and disadvantages in many experiments, in general, Adam has the smallest RMSE when it is used as an adaptive optimization algorithm. Therefore, in this experiment we use the Adam Optimizer for gradient optimization to obtain the optimal model parameters.

After training the BiLSTM model, we input the feature values extracted from the BiLSTM model into the XGBoost model. Similarly, the hyperparameters of the XGBoost model are iterated from the cross-validation set using the grid search method, and the final iterated parameter settings are shown in Table 3.

**Table 3  XGBoost parameter settings.**

| Parameter name | Setting result |
|---|---|
| learning_rate | 0.05 |
| n_estimators | 700 |
| max_depth | 6 |
| min_child_weight | 1 |
| Subsample | 0.8 |
| colsample_bytree | 0.8 |

In the model testing phase, the processed data6 is input into the already trained BiLSTM-XGBoost model to predict the time series with a step size of 50, and the predicted value is compared with the target value, and the fitting results of the three experiments are shown in Fig. 10, where the red curve represents the target value of the model, and the blue curve represents the predicted value of the model.

It can be seen that the prediction results of the BiLSTM-XGBoost model for the trend of future changes in capacitance values are somewhat interpretable and successfully predict a portion of the future trend of capacitance changes, *e.g.*, in the curves of Fig. 10A, where the number of driving times is about 590 times or so and about 780 times or so, and in the curve of Fig. 10B where the number of driving times is about 420 times or so. The prediction models both predict that the driving voltage will steeply drop in the future. Although the actual steep drop in voltage occurs later than predicted, this phenomenon still has a positive significance for correctly evaluating the state of the device compared to conventional reliability assessment methods.

Therefore, we can assume that the model has successfully captured the key features in the data in this dataset, rather than just simply fitting a curve. Furthermore, model interpretability, enhanced by the attention mechanism, provides valuable insight into the model's focus, a crucial aspect in applying AI for engineering predictions (*Hu et al., 2024*; *Xie et al., 2024*) shown in Fig. 11, visualizing the attention weights reveals that the model tends to assign higher attention scores to earlier time steps, especially those with sudden fluctuations or significant changes. These regions often correspond to early signs of system degradation. In addition, the BiLSTM-XGBoost model fits the trend of the sample parameters better, and its output results are more consistent with the target values. Therefore, the model calculation results are highly credible as predictive values, providing valuable reference data for the later stages of reliability and life testing and evaluation (*Hu et al., 2024*).

According to the output of the BiLSTM-XGBoost model, we can make a reliability assessment of the device, following the previous definition of the operating state of the DMF device, if the device is judged to be in the normal wear and tear period, it means that the device is in the early part of its life cycle, the performance of the device is slowly declining, but does not affect the normal use of the device, so we do not need to take corresponding measures, we only need to continue to monitor the device Therefore, we do

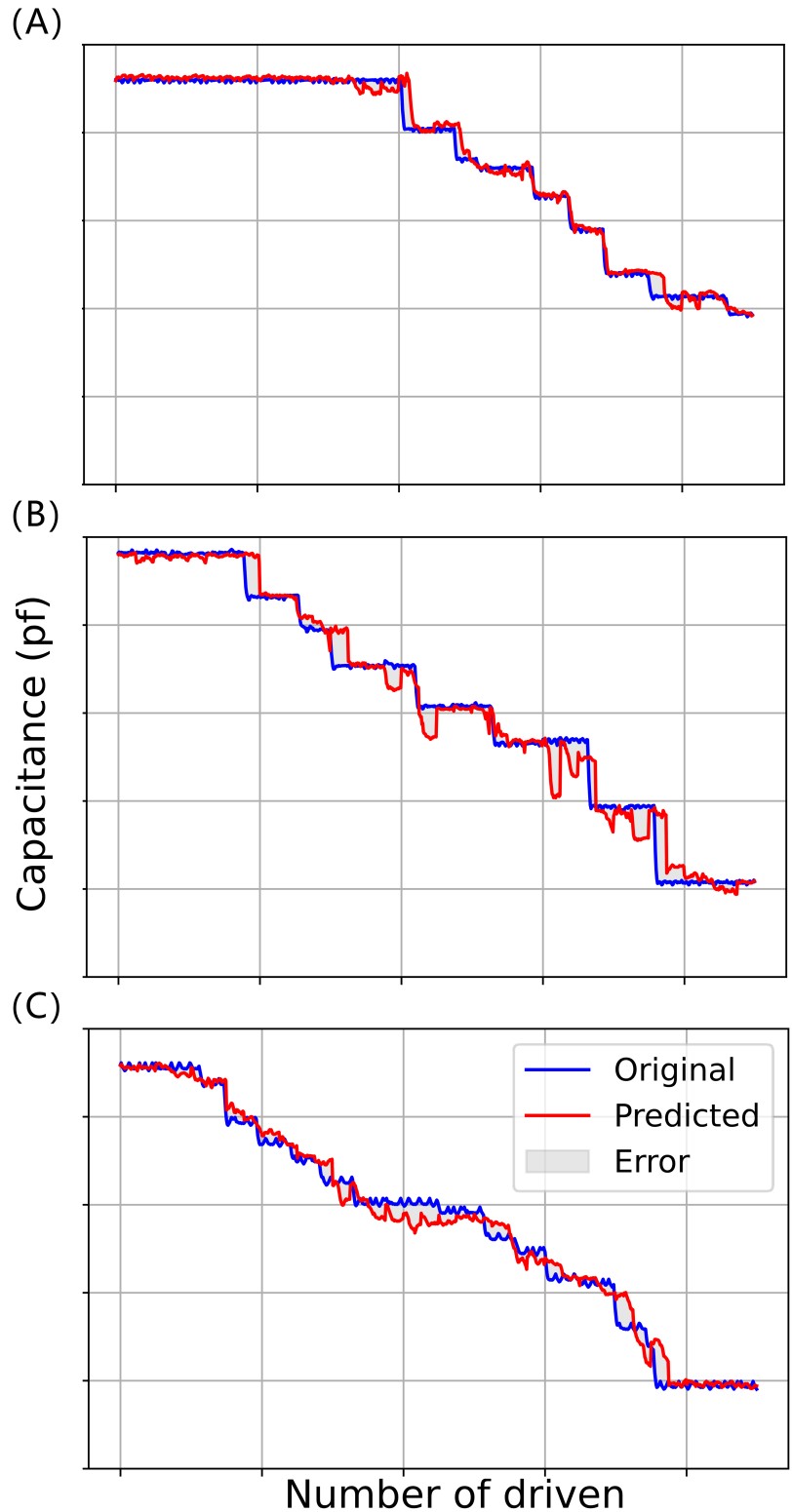

**Figure 10 Model fitting results for three driving voltages: (A) 45 V driving voltage; (B) 70 V driving voltage; (C) 90 V drive voltage.**
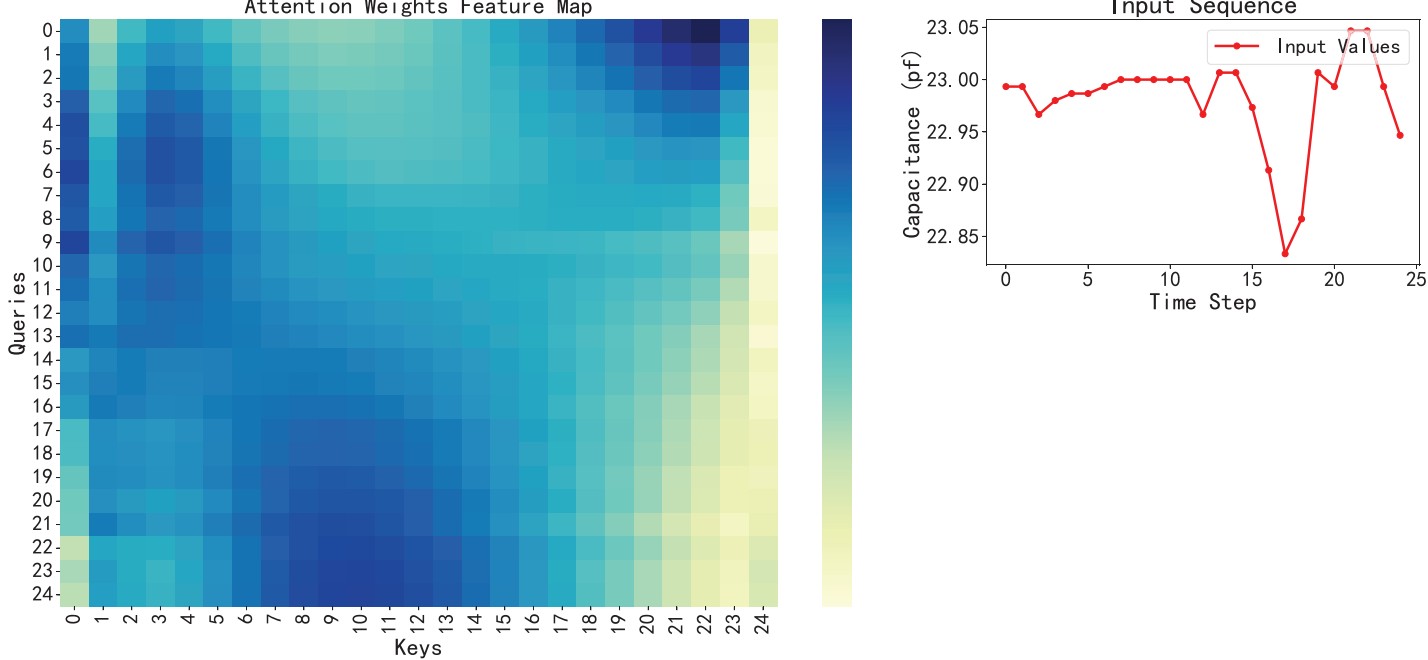

**Figure 11 Visualization of attention weights feature map with the corresponding input sequence.**

not need to take corresponding measures, just need to continuously monitor the device status; if the device is judged to be in the decline period, when the device performance decreases at an increased rate, then we need to pay closer attention to the status of the device and take some measures, such as lowering the voltage or changing the droplet movement path to prevent it from getting stuck on specific electrodes. At the same time, we need to predict the remaining lifetime of the device in conjunction with the damage period threshold, to replace the device promptly, to minimize the loss caused by the sudden failure of the device.

## Comparative analysis of LSTM-XGBoost model with commonly used timing prediction methods

To further validate the effectiveness and superiority of the BiLSTM-XGBoost combined prediction model proposed in this study, we set up other schemes for comparative analysis with the method proposed in this study. In this experiment, the BiLSTM model, XGBoost model, BiGRU model, and BiGRU-XGBoost model were used for comparison. Among them, the comparison using the BiLSTM model, XGBoost model, and BiLSTM-XGBoost model can prove whether the combined model has a better performance relative to a single model in this experiment. Gated recurrent unit (GRU) is a recurrent neural network that invokes an update gate and reset gate to control the information update and is simpler than the LSTM model (*Chung et al., 2014*). To investigate whether the BiLSTM model has better performance than the BiGRU model in this experiment, we use the same idea to construct the BiGRU model and use the same attention-based architecture for fair comparison.

**Table 4 Comparison of test set model evaluation at 45 V drive voltage.**

| Model | MAPE | RMSE | MAE | R-squared |
|---|---|---|---|---|
| BiLSTM | 1.6087 | 0.4147 | 0.2422 | 0.9934 |
| BiLSTM-XGBoost | 1.6464 | 0.3667 | 0.2557 | 0.9949 |
| XGBoost | 1.8783 | 0.5055 | 0.2957 | 0.9903 |
| BiGRU | 1.5426 | 0.3685 | 0.2303 | 0.9948 |
| BiGRU-XGBoost | 1.7197 | 0.4172 | 0.2651 | 0.9933 |

Note:
Mean absolute percentage error (MAPE); root mean square error (RMSE); mean absolute error (MAE); R-squared factor (R-squared).

To accurately assess the predictive performance of the model, we use four key metrics to make a comprehensive assessment of the reliability of the model, and these assessments are as follows:

Mean absolute percentage error (MAPE):

$$e_{MAPE} = \frac{100\%}{m} \sum_{i=1}^{n} \left| \frac{x(i) - y(i)}{y(i)} \right| \tag{10}$$

Root mean square error (RMSE):

$$e_{RMSE} = \sqrt{\frac{1}{m} \sum_{i=1}^{n} (x(i) - y(i))^2} \tag{11}$$

Mean absolute error (MAE):

$$e_{MAE} = \frac{1}{m} \sum_{i=1}^{n} |y(i) - x(i)| \tag{12}$$

R-squared factor (R-squared):

$$R^2 = 1 - \frac{\sum_{i=1}^{n} (x(i) - y(i))^2}{\sum_{i=1}^{n} (y(i) - \bar{y}(i))^2} \tag{13}$$

Equations (10), (11), (12), and (13) represent the training set size, the first prediction value in the training set, and the label of the first data in the training set. Based on the above formulas, the results of the comparative analysis of the comprehensive evaluation of BiLSTM, BiLSTM-XGBoost, XGBoost, BiGRU, and BiGRU-XGBoost models are shown in Tables 4, 5, and 6.

Evaluation results across the tested driving voltages (Tables 4, 5, and 6) reveal distinct model performances. At the higher voltages of 70 V and 90 V, the BiLSTM-XGBoost model consistently demonstrates superior overall prediction performance. However, the scenario at 45 V presents a more complex picture. At this voltage, both the BiGRU and BiLSTM-XGBoost models exhibit competitive performance, each showing respective advantages in different evaluation metrics. Notably, at this same 45 V, the BiGRU-XGBoost model's performance is unexpectedly poorer than that of the standalone BiGRU model, indicating a specific degradation for the hybrid approach under this

**Table 5 Comparison of test set model evaluation at 70 V drive voltage.**

| Model | MAPE | RMSE | MAE | R-squared |
|---|---|---|---|---|
| BiLSTM | 6.7371 | 1.0729 | 0.6963 | 0.9724 |
| BiLSTM-XGBoost | 4.5398 | 0.6897 | 0.4833 | 0.9864 |
| XGBoost | 5.0226 | 0.8664 | 0.6070 | 0.9820 |
| BiGRU | 5.7105 | 1.0623 | 0.6401 | 0.9730 |
| BiGRU-XGBoost | 6.2097 | 1.0173 | 0.6376 | 0.9752 |

Note:
Mean absolute percentage error (MAPE); root mean square error (RMSE); mean absolute error (MAE); R-squared factor (R-squared).

**Table 6 Comparison of test set model evaluation at 90 V drive voltage.**

| Model | MAPE | RMSE | MAE | R-squared |
|---|---|---|---|---|
| BiLSTM | 4.4718 | 0.6996 | 0.4740 | 0.9860 |
| BiLSTM-XGBoost | 4.1003 | 0.6181 | 0.4424 | 0.9891 |
| XGBoost | 4.2479 | 0.6711 | 0.4638 | 0.9871 |
| BiGRU | 4.5398 | 0.6897 | 0.4833 | 0.9864 |
| BiGRU-XGBoost | 4.3944 | 0.6781 | 0.4690 | 0.9868 |

Note:
Mean absolute percentage error (MAPE); root mean square error (RMSE); mean absolute error (MAE); R-squared factor (R-squared).

condition. Despite the particular characteristics observed at 45 V, the BiLSTM-XGBoost model generally achieves the most favorable prediction results across the full range of tested voltages.

For the experimental results, the overall test accuracy at 45 V driving voltage is higher compared to 70 V, 90 V driving voltage, and it can be assumed that the difficulty of predicting 45 V driving voltage is lower, but the BiGRU model performs better under the prediction task for this driving voltage, which we believe is mainly due to its architectural simplification, which allows it to have higher accuracy in simple prediction tasks. However, this also means that the BiGRU model has a relatively weak feature extraction capability, which does not allow XGBoost to learn deeper features from the extracted ones. As a result, the prediction metrics of the BiGRU model are lower than those of the BiLSTM-XGBoost model in both the 70 and 90 V driving voltage prediction tasks. Comprehensively comparing the evaluation results of the BiLSTM model, the XGBoost model and the BiLSTM-XGBoost model, it can be seen that the predictive indexes of the BiLSTM-XGBoost model are better than the single model as a whole under the Stacking's methodology, which provides more informative data for the device's subsequent reliability evaluation and lifetime prediction.

To verify the effectiveness of each part of the BiLSTM pre-trained model, a series of ablation experiments were conducted, and the specific results are shown in Tables 7, 8, and 9. The experiments show that the prediction model using both the two-layer BiLSTM model and the Attention layer achieves the best results in the prediction tasks for all three driving voltages. This indicates that with sufficient data, the models can extract more

**Table 7 Evaluation of ablation experiment results at 45 V driving voltage.**

| Model | MAPE | RMSE | MAE | R-squared |
|---|---|---|---|---|
| Attention+BiLSTM (3-layer) | 2.0548 | 0.4843 | 0.3111 | 0.9910 |
| Attention+BiLSTM (2-layer) | 1.6464 | 0.3667 | 0.2557 | 0.9949 |
| Attention+BiLSTM (1-layer) | 2.0464 | 0.4932 | 0.3276 | 0.9907 |
| BiLSTM (2-layer) | 1.7825 | 0.4617 | 0.2834 | 0.9919 |
| BiLSTM (1-layer) | 2.0077 | 0.4676 | 0.3205 | 0.9917 |
| Attention (2-layer) | 1.9823 | 0.4514 | 0.3172 | 0.9922 |
| Attention (1-layer) | 1.9777 | 0.4330 | 0.2978 | 0.9928 |
| LSTM (2-layer) | 2.3824 | 0.5456 | 0.3700 | 0.9886 |
| LSTM (1-layer) | 2.3244 | 0.5829 | 0.3748 | 0.9870 |

**Note:**
Mean absolute percentage error (MAPE); root mean square error (RMSE); mean absolute error (MAE); R-squared factor (R-squared).

**Table 8 Evaluation of ablation experiment results at 70 V driving voltage.**

| Model | MAPE | RMSE | MAE | R-squared |
|---|---|---|---|---|
| Attention+BiLSTM (3-layer) | 6.0085 | 1.0204 | 0.6387 | 0.9751 |
| Attention+BiLSTM (2-layer) | 4.5398 | 0.6897 | 0.4833 | 0.9864 |
| Attention+BiLSTM (1-layer) | 6.9253 | 1.1823 | 0.7266 | 0.9665 |
| BiLSTM (2-layer) | 5.6568 | 1.0486 | 0.6103 | 0.9737 |
| BiLSTM (1-layer) | 5.0226 | 0.8664 | 0.6070 | 0.9820 |
| Attention (2-layer) | 7.3565 | 1.0842 | 0.7621 | 0.9718 |
| Attention (1-layer) | 7.1298 | 1.2212 | 0.7776 | 0.9643 |
| LSTM (2-layer) | 6.6678 | 1.0655 | 0.6818 | 0.9728 |
| LSTM (1-layer) | 6.6789 | 1.0847 | 0.6591 | 0.9718 |

**Note:**
Mean absolute percentage error (MAPE); root mean square error (RMSE); mean absolute error (MAE); R-squared factor (R-squared).

**Table 9 Evaluation of ablation experiment results at 90 V driving voltage.**

| Model | MAPE | RMSE | MAE | R-squared |
|---|---|---|---|---|
| Attention+BiLSTM (3-layer) | 4.5885 | 0.6549 | 0.4842 | 0.9877 |
| Attention+BiLSTM (2-layer) | 4.1003 | 0.6181 | 0.4424 | 0.9891 |
| Attention+BiLSTM (1-layer) | 4.8613 | 0.7314 | 0.5315 | 0.9847 |
| BiLSTM (2-layer) | 4.7629 | 0.7369 | 0.5186 | 0.9845 |
| BiLSTM (1-layer) | 4.1646 | 0.6532 | 0.4515 | 0.9878 |
| Attention (2-layer) | 5.6472 | 0.8201 | 0.5993 | 0.9808 |
| Attention (1-layer) | 5.6194 | 0.8279 | 0.6040 | 0.9804 |
| LSTM (2-layer) | 4.6063 | 0.6988 | 0.5002 | 0.9860 |
| LSTM (1-layer) | 4.7337 | 0.7509 | 0.5246 | 0.9839 |

**Note:**
Mean absolute percentage error (MAPE); root mean square error (RMSE); mean absolute error (MAE); R-squared factor (R-squared).

**Table 10 Remaining life prediction of DMF devices at three driving voltages.**

| Voltage | Predicted number of damaged drives | Actual number of damaged drives | Error |
| --- | --- | --- | --- |
| 45 V | 754 | 775 | 2.785% |
| 75 V | 610 | 665 | 8.271% |
| 90 V | 602 | 595 | 1.163% |

generalized features as the complexity of the pre-trained model architecture increases. These features allow XGBoost to predict curves with higher accuracy and reduce the risk of overfitting. Thus, the features extracted by our proposed attention-based BiLSTM architecture enable the XGBoost model to have better performance in the prediction task.

## Failure time prediction

According to our defined prediction flow, the remaining lifetime of the EWOD device is predicted when it is in its decline period. This prediction uses the trend of the device capacitance predicted by the BiLSTM-XGBoost model. It is combined with the DMF device damage-period thresholds defined in the previous section. We define the first time that the device capacitance value falls below the defined damage period threshold as a sign of device damage. The predicted and actual results are shown in Table 10.

From the table, it can be seen that the prediction results using the BiLSTM-XGBoost model are able to predict the occurrence of failures in advance. The model can make predictions of possible future failure times even when the device is operating in the recession period. Regardless of the driving voltage used, there is a small relative error between the prediction results and the actual values, showing high prediction accuracy.

## REACH A VERDICT

The attention-based BiLSTM-XGBoost model constitutes a highly advanced analysis tool for DMF systems. It combines the neural network structure of Attention-based BiLSTM with advanced machine learning techniques from XGBoost. This combination provides a new methodology for reliability assessment and lifetime prediction of DMF systems. It leverages BiLSTM's ability to deeply mine complex patterns and long-term dependencies in time series data. Simultaneously, it utilizes XGBoost's powerful data processing and efficient learning mechanisms. The model not only accurately predicts possible failure time points and types, but also provides an accurate estimation of the system's overall lifetime.

The model's application holds significant value. It provides a scientific platform for research teams and enterprises. This platform can significantly improve the efficiency of operation and maintenance management and enhance prediction accuracy for DMF systems. By predicting device reliability and lifetime, operators can schedule usage more scientifically. This helps reduce unnecessary wear and tear. The study's results can assist research teams in making more accurate decisions during droplet manipulation. This will reduce the occurrence of failures in complex DMF applications (*e.g.*, droplets stuck in a certain path) and improve overall operational efficiency and productivity. Additionally, the

predicted results can serve as input parameters for the control system's droplet path planning algorithm, improving its fault tolerance.

The model's development and application provide valuable theoretical support and practical experience for DMF system research and development. With continuous technological progress and accumulation of data, the model's predictive capability and application range are expected to expand. This will enable it to better meet the demand for reliability and life detection in complex industrial environments.

Future research will continue to explore model optimization. This includes parameter tuning, improving feature selection, and evaluating a wider range of application scenarios. These efforts aim to enhance the model's generalization capability and practicality. Additionally, more types of data inputs will be considered, such as image and sound data. Incorporating these will further enhance the accuracy and efficiency of fault detection.

### Funding
This paper was supported by the Guangzhou Science and Technology Plan (No. 201904010233), Guangdong Province Enterprise Science and Technology Commissioner project under Grant GDKTP2021004400, and Rural Science and technology correspondent project of Zengcheng District, Guangzhou city under Grant 2021B42121631. There was no additional external funding received for this study. The funders had no role in study design, data collection and analysis, decision to publish, or preparation of the manuscript.

### Grant Disclosures
The following grant information was disclosed by the authors:
Guangzhou Science and Technology Plan: 201904010233.
Guangdong Province Enterprise Science and Technology Commissioner Project: GDKTP2021004400.
Rural Science and Technology Correspondent Project of Zengcheng District, Guangzhou city: 2021B42121631.

### Competing Interests
The authors declare that they have no competing interests.

### Author Contributions
- Lifeng He conceived and designed the experiments, analyzed the data, performed the computation work, prepared figures and/or tables, authored or reviewed drafts of the article, and approved the final draft.
- Qili Yang performed the experiments, analyzed the data, performed the computation work, prepared figures and/or tables, and approved the final draft.
- Junxi Chen conceived and designed the experiments, performed the experiments, authored or reviewed drafts of the article, and approved the final draft.

- Wenjing Liu conceived and designed the experiments, authored or reviewed drafts of the article, and approved the final draft.
- Zhijie Luo conceived and designed the experiments, performed the computation work, authored or reviewed drafts of the article, and approved the final draft.

## Data Availability

The relevant code and data are available in the Supplemental Files.

## Supplemental Information

Supplemental information for this article can be found online at http://dx.doi.org/10.7717/peerj-cs.3037#supplemental-information.

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
