# Peer review of "Attention-based BiLSTM-XGBoost model for reliability assessment and lifetime prediction of digital microfluidic systems"

_PeerJ Computer Science, doi:10.7717/peerj-cs.3037_

## Round 0.1 · original submission · Major Revisions

After going through your paper and reviewing the comments made by the reviewers on your paper, i agree with the comments made by the reviewers, that the paper need a lot of improvement. In particular you need to pay attention to the language related changes. Please have somebody read and edit the language for you. The reviewers also provide you a detailed suggestion on how to improve the paper. The paper needs more work, but it doable. Based on the reviewers' comments, if you choose to modify, please undertake the revision and submit it as a revised paper.

**Language Note:** The review process has identified that the English language must be improved. PeerJ can provide language editing services - please contact us at [email protected] for pricing (be sure to provide your manuscript number and title). Alternatively, you should make your own arrangements to improve the language quality and provide details in your response letter. – PeerJ Staff

Reviewer 1 ·

Basic reporting

This paper proposes a novel prediction model that enhances the reliability and lifetime testing processes of digital microfluidic systems. Traditional methods have shortcomings due to their heavy reliance on real-time monitoring data, resulting in delays and inefficiencies in evaluating complex droplet behaviors. To address these issues, the authors introduce an integrated model that combines Attention-based Bidirectional Long Short-Term Memory (BiLSTM) with eXtreme Gradient Boosting (XGBoost) in a Stacking approach. This model aims to provide an efficient identification of health states and accurate predictions of failure times for digital microfluidic devices. I would suggest revisions as below:
1. How does the performance of the proposed BiLSTM-XGBoost model quantitatively compare to other advanced machine learning models beyond those mentioned (like BiGRU), particularly in diverse operational conditions of digital microfluidic systems? Are there scenarios where it may underperform?
2. some references about xgboost and explainable AI to enhance the litterature review: Explainable AI models for predicting drop coalescence in microfluidics device; Explainable AI model for predicting equivalent viscous damping in dual frame-wall resilient system
3. Is the BiLSTM-XGBoost model adaptable to other types of microfluidic systems or different domains altogether? How might the model's structure change if applied to other fields, and what limitations might arise?
4. Can the authors provide more detail on how the attention mechanism impacts the model's performance? Specifically, what features or aspects of the data does the model prioritize, and how was this selectivity validated?

Experimental design

see above

Validity of the findings

see above

Additional comments

n/a

Reviewer 3 ·

Basic reporting

detailed report is attached.

Experimental design

detailed report is attached.

Validity of the findings

detailed report is attached.

Additional comments

detailed report is attached.

Annotated reviews are not available for download in order to protect the identity of reviewers who chose to remain anonymous.

---

## Round 0.2 · accepted · Accept

Thanks for undertaking a thorough revision of the paper.

Reviewer 1 ·

Basic reporting

the paper has been greatly improved

Experimental design

the paper has been greatly improved

Validity of the findings

the paper has been greatly improved

Additional comments

the paper has been greatly improved